# The Impact of Transformative Tourism Experiences on Prosocial Behaviors of College Students: Multiple Chain Mediating Effects of Dispositional Awe and Social Connectedness

Ying Li *, Xuan Wu, Yu-Jie Chu and Ya-Jun Guo

Department of Tourism Management, Northwest University, Xi'an 710100, China
* Correspondence: liying@nwu.edu.cn

**Abstract:** As sustainable behaviors that contribute to the development of human society, prosocial behaviors are an important part of the moral cultivation of college students and have attracted the growing attention of higher education in recent years. It has been indicated by previous studies that transformative tourism experiences can have a profound impact on individual prosocial behaviors. Therefore, how transformative tourism experiences play a role in strengthening college students' prosocial behaviors has become a topic worthy of note. Based on the self-determination theory, the awe prototype theory, and the transformative tourism research, this study constructed a mechanism model of the transformative tourism experiences affecting college students' prosocial behaviors. Four hundred and fifty-four valid questionnaires were collected through questionnaire surveys, with the structural equation model and bootstrap analysis method used for the empirical test. The results showed that transformative tourism experiences became one of the important ways to strengthen college students' prosocial behaviors and that the multiple chain intermediary effects of awe and social connectedness between transformative tourism experiences and the college students' prosocial behaviors were significant. This study provided a new way to cultivate college students' prosocial behaviors and promote the sustainable development of human society, and it provided a theoretical basis for the social education function played by research tourism in higher education.

**Keywords:** transformative tourism experiences; dispositional awe; social connectedness; prosocial behaviors of college students; chain mediation

## 1. Introduction

Statistics from the Chinese Ministry of Education showed that there were a total of 9.09 million fresh graduates (including general and vocational undergraduate and post-graduate students) in 2021, which was expected to rise to 10.76 million in 2022 [1]. The huge number of university graduates should preliminarily complete the transformation from "natural humans", in the sense of biological organisms, to "social humans" who take certain social responsibilities and adapt to social life at the university stage. If they are not sufficiently socialized, they will show negative symptoms such as passivity, resistance, isolation, and maladjustment when they enter society or a group after graduation [2], thereby causing a series of personal and social problems. Prosocial behaviors, referring to the positive behaviors that voluntarily benefit others and society, such as helping, cooperating, and comforting [3,4], are an important aspect of positive psychological qualities, as well as a significant part of a sound personality and individual socialization [5,6]. Currently, strengthening the cultivation of prosocial behaviors among college students has become the focus of educational reforms in many places of the world [5]. As the main force of future social construction, college students' prosocial behaviors affect individuals and the whole society; they are not only a guarantee for college students to build a harmonious interpersonal network, ensure physical and mental health, and promote all-round develop-

ment [7], but also a cornerstone of achieving the sustainable development of human society by enhancing college students' sense of social responsibility and moral behaviors [8].

The prosocial behaviors of college students are influenced by various factors, and the existing studies mainly focus on the internal factors (individual characteristics, emotions, cognition, etc.) and the external factors (the usual environments such as school and family situations, etc.) [9]. However, in recent years, the research on the influence of unusual environments, such as tourism experiences, on college students' prosocial behaviors has been on the increase [6]. Nowadays, tourism has become a common way of life, learning, and growth for college students. With its social and educational functions becoming increasingly prominent, transformative tourism experiences, the certain type of tourism experiences that inspire a transformative process and a transformative outcome for a tourist, have gradually entered the research horizons of scholars. In addition, transformative tourism experiences are mostly found in volunteer tourism, backpacker tourism [10], cycling tourism [11], dark tourism [12], red tourism [13] and other niche forms of tourism [14], which engage tourists in "conversations involving the assessment of beliefs, emotions, and values" and then trigger the profound self-change and behavioral change in the tourists [15]. As an illustration, heightening the willingness of tourists to help others and society is among the manifestations of the behavioral change [16]. Obviously, transformative tourism experiences act on individual prosocial behaviors, and compared to other types of tourism experiences, the transformative effects of transformative tourism experiences are profound and continuous [15], enabling their prosocial aftereffects to remain significant even after the tourists return home.

Dispositional awe refers to the propensity of individuals to feel awe in daily life, which reflects individual differences in awe perception and is characterized by relative stability [17]. It can act effectively on prosocial behaviors over time and is mostly used as an independent variable for prosocial behaviors in existing studies because it is not easily changed by external influences [18]. However, the powerful transformative effects of transformative tourism experiences can deeply affect individuals from the outside to the inside, thus enhancing the level of individual dispositional awe and ultimately promoting prosocial behaviors. Meanwhile, social connectedness, as an attribute of the self, reflects the perceptions of individuals of their interpersonal closeness in social life [19] and is considered as an important endogenous mediator in the mechanism of the awe–prosocial effect [20]. Therefore, there is a conductive relationship between transformative tourism experiences, dispositional awe, social connectedness, and prosocial behaviors. However, most of the existing literature on transformative tourism experiences is conceptual and theoretical [21–23], and the specific pathways that influence the post-tour prosocial behaviors of individuals and the role of certain self-changes in individuals, including dispositional awe and social connectedness, remain to be clarified.

Self-determination theory (SDT) states that individuals have a self-determined tendency to direct their self-development and inner growth [24] and that their autonomous behaviors are better motivated by three major sources of motivation: emotion, drive, and intrinsic need. Therefore, from the logical framework of "environmental stimuli (transformative tourism experiences)—emotional motivation (dispositional awe) & drive and intrinsic need motivations (social connectedness)—self-determined behaviors (prosocial behaviors)", the structural equation model studying the effects of transformative tourism experiences on the prosocial behaviors of college students will be constructed and empirically tested with dispositional awe and social connectedness as mediating factors; this is not only of benefit to the complementing of the above research gaps but also of guiding significance to the achievement of the social education function of tourism and the realization of a sustainable society.

## 2. Conceptual Framework and Research Hypotheses

*2.1. Transformative Tourism Experiences and Prosocial Behaviors*

SDT suggests that self-determined behaviors begin with information input from the environment and the structure of the individual's needs. Transformative tourism places individuals in a completely new environment, where the emergence of new resources and challenges creates external stimuli that prompt individuals to reflect and change on their own, making them more tolerant [25,26]. Therefore, transformative tourism experiences can strengthen tourists' prosocial behaviors by changing their perceptions of themselves and the environment [27]. In the new environment of transformative tourism, individuals experience novelty and harvest by enjoying stunning mountains and rivers, feeling deep contact with nature and society, and meeting new friends they could not have met before, or they experience hardship and pain by visiting black tourist sites and facing unexpected situations. These experiences are new meaningful resources and challenges for college tourists, and these external stimuli will prompt college tourists to more deeply realize the importance of a good life and social harmony based on either positive or negative experiences, so as to establish their outlooks on the world, life, and the values of caring for others and society [15], and they will eventually promote the realization of the long-lasting prosocial behaviors of the college tourists after the tour from the bottom of their hearts. Some studies have shown that individuals who experienced transformative tourism have the potential to positively impact their communities upon return [28]; for example, "couch surfers" tend to build a better world in which humans around the world will become brothers or sisters and live in harmony with people and even strangers [29]. However, it has also been found that transformative tourism experiences can negatively affect the prosocial behaviors of tourists after returning home; as an illustration, the excessive consumerism and materialism felt during the trip may cause tourists to exhibit a detachment from society and others [30]. Therefore, the relationship between transformative tourism experiences and prosocial behaviors remains to be studied in depth. Combined with the Chinese ethics based on the family and home, as well as the fact that public security is good and dark tourism is minor in China, this study believes that the more intense the transformative tourism experiences are, the more insights tourists will gain from the environment, which leads to an increased sense of helping others and society and then profoundly and persistently enhances their prosocial behaviors, even after the tourists return home. This results in hypothesis 1.

**H1:** *Transformative tourism experiences have a significantly positive impact on college tourists' prosocial behaviors.*

*2.2. Transformative Tourism Experiences, Dispositional Awe, and Social Connectedness*

Most research on awe in tourism has focused on state awe during the tour, but dispositional awe, as a determinant of the daily level of individual awe, is more capable of providing lasting emotional motivation for individual self-determined prosocial behaviors [31]. Therefore, this study will focus on dispositional awe in college tourists. Transformative tourism can help tourists develop a more open worldview [25], and openness is a common influence on dispositional awe among college students [32]. As a result, the challenging environment of transformative tourism experiences would be more conducive to enhancing the individual's dispositional awe than other types of tourism experiences. It has been suggested that transformative tourism experiences cause tourists to shift their previous attitudes and perceptions and become appreciative and aware of the vast environment and wide world [15], making tourists more inclusive and open and improving their propensity to perceive awe. Furthermore, the relative stability of dispositional awe refers to the fact that dispositional awe is generally not easily changed, but once changed, it is re-stabilized and maintained at a new level. Thus, transformative tourism experiences can have a dramatic intrapersonal change that enables tourists to discover and feel many tangible and intangible forces even after returning to their hometowns, which will continuously

and positively promote the dispositional awe of college students. Therefore, hypothesis 2 is proposed.

**H2:** *Transformative tourism experiences have a significantly positive impact on college tourists' dispositional awe.*

Social connectedness represents the basic need of individuals to connect with others and to belong to other people or groups and constitutes the foundation of human social interaction [33]. As the subjective feelings and self-perceptions of an individual's intimate relationship with the society, social connectedness can be changed by stimuli in the travel context. For example, the interactions between tourists can affect tourists' social connectedness [34]. One of the differences between transformative tourism experiences and other types of tourism experiences lies in the interactivity with others during the journey [14]. Previous studies have shown that backpackers gain social capital by accepting tangible (social networks and connections) and intangible (shared norms and ideas) changes, ultimately improving the tolerance and understanding of others [35], promoting intimacy, and shortening cognitive distance from each other. This means that transformative and awakening travel will help tourists achieve a higher sense of connectedness [35]. Therefore, through transformative tourism experiences, tourists will be more willing and likely to accept others and to be closer to the community, thus enhancing their sense of social connectedness. Evidently, the new social relationships established based on interactions in the context of transformative tourism can strengthen college tourists' sense of social connectedness. Accordingly, hypothesis 3 is proposed.

**H3:** *Transformative tourism experiences have a significantly positive impact on college tourists' social connectedness.*

### 2.3. Dispositional Awe, Social Connectedness, and Prosocial Behaviors

SDT indicates that emotion is one of the motivations for self-determined behaviors. Dispositional awe, as an important determinant of the daily level of an individual's awe, can provide sustained emotional motivation for autonomous prosocial behaviors. High dispositional awe, which means the greater frequency and intensity of induced awe emotions, can strongly increase an individual's sense of belonging to a larger group, decrease egocentric tendencies, increase collective identification [36,37], and finally lead to altruistic motivations. Accordingly, individuals with high levels of dispositional awe generally have lower self-importance, which makes them more adept at turning their attention to a larger collective group [38] and at actively developing more prosocial behaviors based on greater concern for the interests and perceptions of others. So far, positive emotions have been studied to promote positive behaviors and inhibit negative behaviors in tourists [39]. Dispositional awe, as one of the dispositional positive emotions, has been found to have a positive prediction effect on the individual's prosocial tendencies or behaviors [40,41]. Hence, the higher the college tourists' level of dispositional awe, the stronger the reinforcement effect on their prosocial behaviors will be. Due to the relative stability of dispositional awe, this reinforcement effect will continue to act on the post-tour prosocial behaviors of college tourists. Therefore, hypothesis 4 is proposed.

**H4:** *Dispositional awe has a significantly positive impact on college tourists' prosocial behaviors.*

According to SDT, on the one hand, social connectedness creates a strong sense of connection with society and with others [9], which will become a driving force that makes individuals more willing to engage in society. On the other hand, under the effect of a sense of belonging and intimacy, the desire for interdependence between self and others will be stimulated [42], making individuals form an intrinsic need to be close to society and others. Therefore, individuals will have a stronger tendency for prosocial behaviors under the dual motivation of drive and intrinsic need. The higher the degree of social connectedness, the friendlier the individual's attitude toward others and society will be. Otherwise, individuals will be more prone to social detachment and social discomfort [19]. Researchers

have discovered that social connectedness has a significantly positive impact on individual donation behavior [43] and that college students with high social connectedness are more willing to accept prosocial behavioral goals [44]. It can be inferred that social connectedness positively affects the prosocial behaviors of college tourists even after returning home. This is because social connectedness is classified as a type of self-attribute, which can lastingly influence individual behaviors. Therefore, hypothesis 5 is proposed.

**H5:** *Social connectedness has a significantly positive impact on college tourists' prosocial behaviors.*

*2.4. Multiple Chain Mediating Effects of Dispositional Awe and Social Connectedness*

The positive effect of transformative tourism experiences on tourists' openness helps to enhance the individual's dispositional awe, and under the effect of dispositional awe, individuals will subconsciously pay more attention to the collective group than to themselves [38], thereby consistently reinforcing their prosocial behaviors. Previous studies have also shown that the environments in which individuals stay can influence their dispositional awe and act on their subsequent behavior [45]. Therefore, dispositional awe may play a role between tourism experiences and the tourists' behavior. In addition, according to SDT, transformative tourism experiences can help to enhance the dispositional awe of college tourists, which will induce more awe of others and society, reduce their "small self", enhance their sense of community, strengthen their prosocial behaviors, and eventually complete the transformation process from environmental stimuli to emotional motivations and ultimately to self-determined behaviors. Therefore, hypothesis 6 is proposed.

**H6:** *Dispositional awe mediates the relation between transformative tourism experiences and college tourists' prosocial behaviors.*

Social connectedness emphasizes the perceptions of individuals of their social relationships, and transformative tourism experiences with interactive characteristics can help college tourists establish new social relationships and thus increase their level of social connectedness. High social connectedness gives college tourists a sense of connection with society and others and a desire to rely on the outside world, both of which long-lastingly reinforce college tourists' prosocial behaviors from the dual perspectives of drive and intrinsic need, thus forming a transformation mechanism of SDT from environmental stimuli to drive and intrinsic need motivations and ultimately to self-determined behaviors. Researchers have found that sports tourism experiences affect individual behavior by influencing social connectedness [46]. Therefore, social connectedness plays a mediating role between tourism experiences and individual behaviors, which leads to hypothesis 7. Furthermore, dispositional awe makes individuals more likely to blur the conscious boundary between self and the external world, which is precisely the characteristic of social connectedness [19]. Consequently, the awe emotional tendency can effectively improve an individual's social connectedness [47], and it is easier to form a strong sense of connectedness with other beings with higher dispositional awe [48]. Therefore, hypothesis 8 is proposed. In summary, based on SDT and the logical framework of "environmental stimuli (transformative tourism experiences)—emotional motivation (dispositional awe) & drive and intrinsic need motivations (social connectedness)—self-determined behaviors (prosocial behaviors)", this study concludes that transformative tourism experiences contribute to dispositional awe, and dispositional awe promotes social connectedness, ultimately leading to long-lasting prosocial behaviors in college tourists. Based on the above hypotheses, the conceptual model of this study is proposed as below (see Figure 1).

**H7:** *Social connectedness mediates the relation between transformative tourism experiences and college tourists' prosocial behaviors.*

**H8:** *Dispositional awe has a significantly positive impact on social connectedness.*

**H9:** *Dispositional awe and social connectedness play a chain mediating role between transformative tourism experiences and college tourists' prosocial behaviors.*

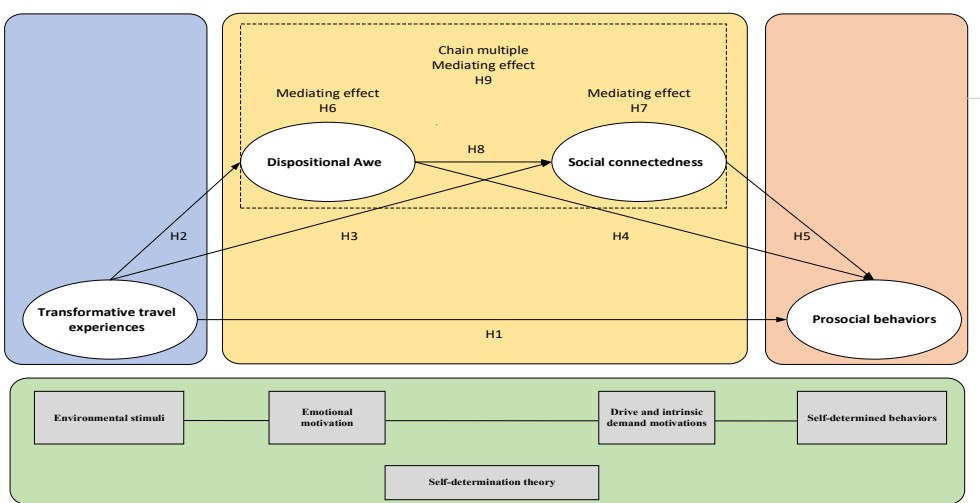

**Figure 1.** Research model.

## 3. Methodology

### 3.1. Measurements

To improve the reliability and validity of conceptual operationalization, the measurements of 4 variables were based on the maturity scales used and tested in the existing literature, especially in accordance with the research situation of this study. Transformative tourism experiences were measured using 8 items adapted from Tasci and Godovykh [15]. Dispositional awe was measured using 5 items adapted from Rui Dong [31]. Social connectedness was measured using 8 items adapted from Lee and Robbins [19]. Prosocial behaviors were measured using 23 items adapted from Wen-jun Cong [9]. The four conceptual scales were all measured on a five-point Likert scale. To make the scales of dispositional awe, social connectedness, and prosocial behaviors fit the long-term reform theme of the study, the relevant questions were adapted to express the meaning of post-tour and transformation. Finally, ten college tourists with transformative tourism experiences were invited to finish a pilot test to correct the unclear or ambiguous expressions and form a formal survey questionnaire.

The questionnaire of this study included two parts: basic personal information and principal variable measurement scales. To be specific, the basic personal information consisted of 1 screening item designed by referring to the study of Tasci and Godovykh to ensure that the respondents had a transformative travel experience ("Please recall a travel that changed your worldview, attitude, behavior, personality, beliefs, or vision of self and to what extent did it change you?"), 1 item representing the demographic sociological attribute (gender), and 3 items representing a specific travel context (travel type, travel duration, and travel mode) that could help the respondents better recall their transformative travel experiences. According to the existing quantitative research on transformative tourism experiences, this study adopted a post-tour and self-report approach to collect data for the purpose of verifying the long-lasting effects of transformative tourism experiences and enhancing the research significance of prosocial behaviors more generally.

### 3.2. Sample and Procedure

To ensure the reliability and validity of the research questionnaire, on-campus field pre-survey and questionnaire optimization were conducted with the help of a platform named Questionnaire Star before carrying out the final formal survey, and a total of 125 valid pre-survey questionnaires were collected. The initial scale items were purified using principal component analysis, and the criteria for removing items included: (1) items with loads smaller than 0.5 on a single factor; (2) items with loads larger than 0.4 on multiple factors; (3) items with negative contributions to Cronbach's $\alpha$; and (4) items with corrected item-total correlations smaller than 0.5. Then, based on the feedback from the pre-survey, the

formulation of the items was optimized. Finally, the purified scales contained a total of 29 items. The formal surveys were conducted in three periods between 9 November 2021 and 10 January 2022.

Firstly, from 9 to 22 November 2021, a one-on-one questionnaire survey was conducted first among college students with transformative tourism experiences around the authors. Then, a snowball sampling method was used to expand the field of investigation. Afterwards, the questionnaires were distributed among the WeChat group of college students assisting education in poverty-stricken areas, the WeChat group of college students working as youth volunteers, the WeChat group of cycling teachers and students in Shaanxi universities, and the QQ group of cycling college students, resulting in 554 questionnaires collected in the first round of investigation. Secondly, two offline questionnaire surveys were carried out from 23 to 28 November 2021 and from 5 to 10 January 2022. The field surveys were conducted at Xi'an Technological University, Xidian University, Northwest Agriculture & Forestry University, Northwest University, and the like, with 235 questionnaires collected. Both the online and the offline surveys had detailed textual or verbal explanations about the notes on filling in the questionnaire and the definition of transformative tourism experiences. A total of 789 questionnaires were collected during the three periods of formal surveys. The respondents who rated themselves as having had no transformative travel experience were excluded according to the screening item. Given that it would take at least 2 min to finish the questionnaire according to the trial questionnaire, the invalid questionnaires that took less than 2 min to fill in were also excluded. Finally, a total of 454 valid questionnaires were collected, reaching an effective rate of 61.60%. The descriptive statistics of the survey sample were reported in Table 1. On the whole, the population characteristics and the tourism characteristics of the sample were highly representative and could meet the data requirements of the follow-up empirical research.

**Table 1.** Descriptive statistical analysis.

| Feature | Type | Proportion |
|---|---|---|
| Gender | Female | 57.61% |
| | Male | 42.37% |
| Travel type | Backpacker travel | 40.95% |
| | Cycling travel | 5.35% |
| | Volunteer travel | 23.04% |
| | Dark travel | 2.67% |
| | Red travel | 17.70% |
| | Others | 10.29% |
| Travel mode | Traveling alone | 15.02% |
| | Traveling with friends | 48.56% |
| | Traveling with couples | 5.56% |
| | Traveling with family | 12.96% |
| | Traveling with units or associations | 17.90% |
| Travel duration | 1–3 days | 37.45% |
| | 4–7 days | 45.06% |
| | 8–15 days | 12.14% |
| | 16–30 days | 3.50% |
| | Above 30 days | 1.85% |

## 4. Empirical Results and Analysis

### 4.1. Common Method Bias Test

This study used a multi-time survey method to reduce common method bias. However, the data collection approach was relatively unique; so, the data might still have a common method bias. The principal component analysis without rotation (Harman's single-factor test) was used to test the influence of the possible common method bias [49]. The results suggested that the maximum unrotated factor variance was 26.178%, which

was smaller than 50%, indicating that common method bias in this study was within the controllable range.

### 4.2. Confirmatory Factor Analysis

With the help of Spss22.0, the KMO test and the Bartlett test of sphericity were carried out on the measurement model. It turned out that the value of the KMO test was 0.888, and the significance of the Bartlett test of sphericity was 0.000, indicating that the measurement model was suitable for factor analysis. Then, AMOS24.0 software was used to test the measurement model. The results in Table 2 showed that the transformative tourism experiences 5-item 2-factor model, the dispositional awe 4-item single-factor model, the social connectedness 5-item single-factor model, the and prosocial behaviors 15-item 6-factor model all met the standard of CMIN/DF < 3, RMSEA < 0.08, GFI, NFI, IFI, TLI, CFI > 0.9, indicating that each measurement model had a good fitting effect. The overall structural model was further examined and the fitting indices (CMIN = 306.046, CMIN/df = 2.708, GFI = 0.927, NFI = 898, IFI = 933, TLI = 0.919, CFI = 0.933, RMSEA = 0.061) were up to the standard, except NFI, which indicated that the structural model had a good fitting effect.

**Table 2.** Fitting indices of the measurement model.

|  | CMIN | CMIN/DF | GFI | NFI | IFI | TLI | CFI | RMSEA |
|---|---|---|---|---|---|---|---|---|
| Transformative tourism experience 5 items 2 factors model | 11.970 | 2.993 | 0.990 | 0.963 | 0.975 | 0.937 | 0.975 | 0.066 |
| Awe 4 items 1 factor model | 3.307 | 1.654 | 0.966 | 0.966 | 0.998 | 0.995 | 0.998 | 0.038 |
| Social connectedness 5 items 1 factors model | 11.999 | 2.400 | 0.990 | 0.989 | 0.994 | 0.987 | 0.994 | 0.056 |
| Prosocial behavior 15 items 6 factors Model | 220.916 | 2.946 | 0.938 | 0.919 | 0.945 | 0.922 | 0.945 | 0.066 |

In addition, as reported in Table 3, the standardized factor loads of most observation variables in the questionnaire were larger than 0.5. Although there were some factor loads smaller than 0.5, they also met the basic requirement of being larger than 0.4, as suggested by Jie-tai Hou, Zhong-lin Wen and Zi-juan Cheng [50], showing that the factors enjoyed a strong explanatory power for the measurement model. The values of Cronbach's α were all larger than 0.8 except that of the transformative tourism experiences. Hatcher and Stepanski believed that Cronbach's α should not be smaller than 0.55 for social science research [51]. Therefore, the scales enjoyed high reliability and good stability. In terms of content validity, this study fully followed the basic procedure and principles of scale development when designing the questionnaire. The concept scales referred to the maturity scales, and then, professional translation and appropriate revision were made so that the problem of the content validity of the measurement could be avoided. The construction validity was examined in terms of convergent validity and discriminant validity. The values of composite reliability (CR) were all larger than 0.8. In addition, the values of average variance extracted (AVE) were all larger than 0.5 except that of the transformative tourism experiences. According to the study of Fornell and Larcker, an AVE larger than 0.36 was acceptable, and an AVE larger than 0.5 indicated that the structure had good convergent validity [52]. Therefore, it could be concluded that there was a good convergent validity between the items measuring the same variable. Furthermore, the correlation coefficients among the four latent variables in Table 4 were significant; the square root of the AVE of each variable was larger than the correlation coefficient between this variable and the other variables; and the minimum square root of AVE (0.680) was larger than the maximum correlation coefficient between variables (0.552), indicating that the discriminant validity of the scales was favorable.

**Table 3.** Results of the confirmatory factor analysis.

| Constructs and Measurement Items | Factor Loading | T Value | Cronbach's α | CR | AVE |
|---|---|---|---|---|---|
| **Transformative tourism experiences** | | | **0.579** | **0.800** | **0.463** |
| **Novelties & Gains** | | | | | |
| During this travel, I met new people (local people, fellow travelers, new friends, etc.). | 0.427 | - | | | |
| During this travel, I engaged in self-improvement activities (physical activities, wellness, meditation, yoga, etc.). | 0.834 | 5.863 *** | | | |
| During this travel, I took on challenging activities (runs, races, extreme activities, etc.). | 0.593 | 7.081 *** | | | |
| **Toils & Pains** | | | | | |
| During this travel, I faced problems (injures, crime, losses, fraud, etc.). | 0.524 | - | | | |
| During this travel, I witnessed tragedies (visiting tragedy and death related tourism places). | 0.899 | 1.715 ** | | | |
| **Awe** | | | **0.852** | **0.855** | **0.596** |
| After this travel, I see more beauty all around me. | 0.732 | - | | | |
| After this travel, I look for patterns in the objects around me more often. | 0.785 | 15.343 *** | | | |
| After this travel, I have more opportunities to see the beauty of nature. | 0.827 | 15.932 *** | | | |
| After this travel, I am more willing to seek out experiences that challenge my understanding of the world. | 0.739 | 14.520 *** | | | |
| **Social connectedness** | | | **0.868** | **0.873** | **0.583** |
| After this travel, I feel a stronger sense of belonging with people I know. | 0.636 | - | | | |
| After this travel, I have more sense of togetherness with my peers. | 0.863 | 14.601 *** | | | |
| After this travel, I feel closer to others. | 0.839 | 14.352 *** | | | |
| After this travel, I feel a stronger sense of connectedness with the society. | 0.637 | 11.640 *** | | | |
| After this travel, I have a deeper sense of brother/sisterhood with my friends. | 0.810 | 14.016 *** | | | |
| **Prosocial behaviors** | | | **0.851** | **0.948** | **0.555** |
| **Public** | | | | | |
| After this travel, when other people are around, it is easier for me to help needy others. | 0.661 | - | | | |
| After this travel, helping others when I am in the spotlight is when I work better. | 0.543 | 9.061 *** | | | |
| **Emotional** | | | | | |
| After this travel, it is more fulfilling to me when I can comfort someone who is very distressed. | 0.484 | - | | | |
| After this travel, emotional situations make me want to help needy others even more. | 0.877 | 8.054 *** | | | |
| **Dire** | | | | | |
| After this travel, I would be more inclined to help people who are in a real crisis or need. | 0.813 | - | | | |
| After this travel, I would be more inclined to help people who hurt themselves badly. | 0.831 | 19.589 *** | | | |
| After this travel, it is easier for me to help others when they are in a dire situation. | 0.875 | 20.661 *** | | | |
| **Altruism** | | | | | |
| After this travel, I think more than ever that one of the best things about helping others is that it makes me look good. | 0.769 | - | | | |
| After this travel, I believe I should receive more recognition for the time and energy I spend on charity work. | 0.679 | 11.675 *** | | | |
| After this travel, I have more convinced that one of the best things about doing charity work is that it looks good on my resume. | 0.517 | 9.396 *** | | | |
| **Compliant** | | | | | |
| After this travel, when people ask me to help them, I don't hesitate even more. | 0.693 | - | | | |
| After this travel, I help others without hesitation when they ask for it more often. | 0.971 | 9.937 *** | | | |
| **Anonymous** | | | | | |
| After this travel, I am more willing to donate money anonymously. | 0.762 | - | | | |
| After this travel, I would be more inclined to help needy others when they do not know who helped them. | 0.821 | 14.822 *** | | | |
| After this travel, I make anonymous donations more often because they make me feel good. | 0.685 | 13.287 *** | | | |

Note: The "***" indicates $p < 0.001$, "**" indicates $p < 0.01$, two-tailed test.

**Table 4.** Results of the discriminant validity test.

|  | **Mean** | **SD** | **1** | **2** | **3** | **4** |
|---|---|---|---|---|---|---|
| 1. Transformative tourism experiences | 2.770 | 0.674 | **0.680** |  |  |  |
| 2. Awe | 4.296 | 0.597 | 0.125 ** | **0.772** |  |  |
| 3. Social connectedness | 4.014 | 0.631 | 0.145 ** | 0.552 ** | **0.764** |  |
| 4. Prosocial behviors | 3.554 | 0.507 | 0.307 ** | 0.352 ** | 0.416 ** | **0.745** |

Note: The value on the diagonal line is the square root of AVE, and the value under the diagonal line is the correlation coefficient between variables. The "**" indicates $p < 0.01$, two-tailed test.

*4.3. Hypothesis Testing*

4.3.1. Test of Structural Equation Model

This study first tested the direct effects in the hypotheses. The test results of the structural equation model using AMOS 24.0 showed that the transformative tourism experiences had a significantly positive effect on prosocial behaviors ($\beta = 0.329$, $p < 0.01$) and dispositional awe ($\beta = 0.264$, $p < 0.01$). Therefore, H1 and H2 were supported. However, the transformative tourism experiences had no significantly positive effect on social connectedness ($\beta = 0.096$, $p > 0.05$); so, H3 did not hold. Meanwhile, dispositional awe had a significantly positive effect on both prosocial behaviors ($\beta = 0.168$, $p < 0.05$) and social connectedness ($\beta = 0.582$, $p < 0.001$); so, H4 and H8 were supported. In addition, social connectedness had a significantly positive effect on prosocial behaviors ($\beta = 0.315$, $p < 0.001$); so, hypothesis H5 was supported.

In summary, the direct hypothesis test results of this study are reported in Figure 2.

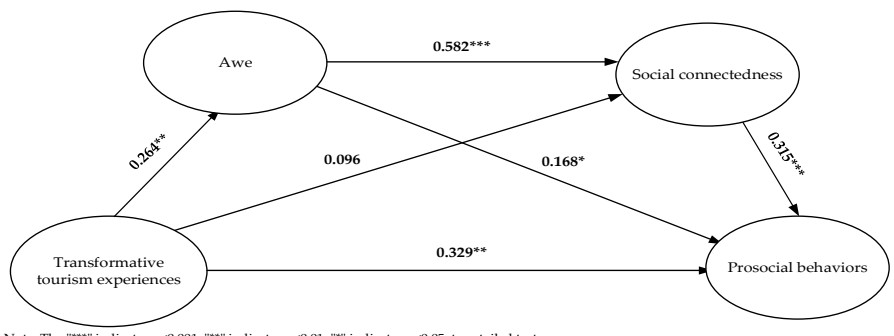

Note: The "***" indicates p<0.001, "**" indicates p<0.01, "*" indicates p<0.05, two-tailed test.

**Figure 2.** Test results of the structural equation model.

4.3.2. Test of Multiple Chain Mediating Effects

In this study, the bootstrap analysis was used to repeat the sampling 5000 times, and Amos 24.0 was adopted to calculate the confidence intervals of total effect, direct effect, and indirect effect under the 95% confidence interval to test the significance of the intermediary effect. The results are reported in Table 5.

**Table 5.** Results of the mediating effect test.

| Effect | Path | β | SE | Bias-Corrected Percentile Method | | |
|---|---|---|---|---|---|---|
|  |  |  |  | **Lower** | **Upper** | **Two-Tailed Test** |
| Total effect | Transformative tourism experiences → Prosocial behaviors | 1.220 | 0.597 | 0.708 | 3.082 | *** |
| Direct effect | Transformative tourism experiences → Prosocial behaviors | 0.888 | 0.538 | 0.454 | 2.237 | *** |
| Indirect effect | Transformative tourism experiences → Dispositional awe → Prosocial behaviors | 0.120 | 0.087 | 0.028 | 0.450 | * |
|  | Transformative tourism experiences → Social connectedness → Prosocial behaviors | 0.081 | 0.065 | 0.009 | 0.332 | * |
|  | Transformative tourism experiences → Dispositional awe → Social connectedness → Prosocial behaviors | 0.131 | 0.069 | 0.053 | 0.436 | ** |

Note: The "***" indicates $p < 0.001$, "**" indicates $p < 0.01$, "*" indicates $p < 0.05$, two-tailed test.

Firstly, the confidence intervals of the total effect and direct effect of "transformative tourism experiences→prosocial behaviors" did not include 0, and the confidence interval of the indirect effect of dispositional awe did not include 0. Therefore, dispositional awe

partially mediated the relation between transformative tourism experiences and prosocial behaviors shown in hypothesis 1, thus supporting hypothesis 6. Secondly, referring to the test of the mediating effect by MacKinnon and Fairchild [53], although the direct effect of transformative tourism experiences on social connectedness was not significant, the bootstrap analysis showed that the confidence interval of the indirect effect of "transformative tourism experiences→social connectedness→prosocial behaviors" did not contain 0. In addition, the product of the path effect values of the independent variable to the mediating variable and of the mediating variable to the dependent variable had the same plus–minus directions as the direct effect value from the independent variable to the dependent variable. Therefore, it could be concluded that social connectedness partially mediated the relation between the transformative tourism experiences and the prosocial behaviors, thereby confirming hypothesis H7. Thirdly, the confidence interval of the indirect effect of "transformative tourism experiences→dispositional awe→social connectedness→prosocial behaviors" did not contain 0, indicating that dispositional awe and social connectedness played a chain mediating role in the effect of the transformative tourism experiences on prosocial behaviors. Therefore, hypothesis H9 was confirmed.

## 5. Discussion

### 5.1. Overview of Key Findings

In addition to reconfirming the two existing ideas that dispositional awe and social connectedness are both key influences on college students' prosocial behaviors, the new findings of this study are as follows.

First, transformative tourism experiences are important for strengthening prosocial behaviors among college students. This study verifies that transformative tourism experiences significantly and positively influence prosocial behaviors. The finding supports the view held by Vidickien, Vilke, Gedminaite-Raudone, and Ateljevic that transformative tourism experiences have prolonged effects on tourists' thoughts and behaviors [54], contribute to individual and collective development, and promote social and environmental justice [55]. In addition, it is also consistent with the conclusion reached by Mkono and Decrop et al. that transformative tourists will actively pay back to their communities [28,29]. According to SDT, when individuals are placed in the context of transformative tourism, the novel and challenging external environment gives them specific motivations of emotion, drive, and intrinsic need and ultimately acts on their self-determined behaviors. Evidently, the environmental stimuli of transformative tourism experiences play a significant role in strengthening tourists' prosocial behaviors.

Second, transformative tourism experiences are essential enhancers of dispositional awe among college students. The results show that transformative tourism experiences significantly and positively influence dispositional awe and social connectedness. There is no direct empirical evidence in previous literature on the relationship between transformative tourism experiences, dispositional awe, and social connectedness. However, Force, Manuel-Navarrete, and Benessaiah found that transformative tourism experiences can act within individuals and improve tourists' openness [25]. Razavi et al. noted that openness is a common influencing factor on dispositional awe [56]. Arguably, transformative tourism experiences can help tourists to develop an open worldview under the "small self" to enhance their dispositional awe by perceiving specific elements, such as natural landscapes, novel cultures, and connections with others.

Third, transformative tourism experiences promote prosocial behaviors of college students by strengthening dispositional awe and social connectedness in turn. To begin with, the empirical finding that dispositional awe and social connectedness play a mediating role between transformative tourism experiences and prosocial behaviors further validates the view of Yu and Liu et al. that dispositional awe and social connectedness act as mediators between environmental experiences and individual behaviors from the perspective of tourism [45,46]. In addition, the insignificantly direct effect of transformative tourism experiences on social connectedness reflects the possibility that there may be

some unknown mechanisms between the two that remain to be included in the research field. This also echoes the empirical finding that the mediating effect of dispositional awe is stronger than that of social connectedness at the same significance level, suggesting that dispositional awe has stronger explanatory power than social connectedness in the effect mechanism of transformative tourism experiences on prosocial behaviors among college students. Furthermore, combined with the positive effect of dispositional awe on social connectedness, this study verifies for the first time that there is a significant chain mediating effect of dispositional awe and social connectedness on the relationship between transformative tourism experiences and college tourists' prosocial behaviors. According to SDT, the self-reflective conditions that transformative tourism experiences create for tourists through environmental stimuli help them develop a more open worldview [25,26], which can enhance the dispositional awe of individuals [56]. Meanwhile, dispositional awe helps to improve the individual's social connectedness [57]. Ultimately, tourists' prosocial behaviors are reinforced lastingly and effectively by the combined effect of three major motivations: emotion, drive, and intrinsic need.

*5.2. Theoretical Contributions and Practical Implications*

The first theoretical contribution is to refine and validate the effect of transformative tourism experiences on the internal psychology and external behaviors of college students. Most of the current studies on transformative tourism experiences focused on conceptual definition and supplementation using qualitative methods, as in those of Soulard, Mcgehee, Stern, and Lamoureux [58]. Only a few empirical researchers, such as Tasci and Godovykh, found that transformative tourism experiences can affect individual attitudes, personalities, values, and behaviors internally and externally [15]. However, they failed to further explore the impact of transformative tourism experiences on some specific types of internal psychology or emotions and external behaviors. By contrast, this study focuses on the aftereffects of transformative tourism experiences on prosocial behaviors and the internal mediators in this mechanism, while expanding the conceptual connotation of transformative tourism experiences. The second theoretical contribution is to enrich the research on multiple types of awe within the tourism field. Most conducted studies on tourist awe only measured state awe; yet, the positive aspects of tourism go far beyond its transient efficacy. Tourism can have profound influences on individuals at the dispositional level. This study provides a theoretical basis for the contribution of transformative tourism experiences to dispositional awe, not only by deepening the research on the impact of tourism experiences on various individual awe types, but also by providing new empirical support for the view held by Yan-zheng Tuo, Bai Chang-hong Bai, and Lin Wang that tourism is about the quality of life and individual growth [59,60].

As for the practical implications, tourism destinations should first increase the supply of transformative tourism experience scenes according to local conditions and give full consideration to how to enhance the dispositional awe and social connectedness of college tourists. Facing the diversified and personalized demands of the tourism market, tourist destinations should design more tourism products, such as volunteer tourism and cycling tourism, that can provide a profound impact for tourists. In the meantime, tourism destinations need to improve the dispositional awe of college tourists by enhancing their openness with the help of diversified attractions and experience projects and strengthening the social connectedness of college tourists by increasing their sense of belonging through warm and harmonious host–guest interactions and collective activities. Secondly, colleges and universities should make reasonable use of the travel context in an unusual environment and integrate transformative tourism activities in multiple scenes and ways into the life and study of college students in the form of a second classroom, social practice, and clubs and organizations, so as to give full play to the social education function of tourism. Currently, the body of college students belongs to "Generation Z", with such identity characteristics as being "only child or few children", "internet natives", strong self-awareness, and weak socialization [61], which makes their level of dispositional awe

and social connection relatively weak and their prosocial tendency low. Accordingly, universities need to educate college students about the importance of transformative tourism experiences for their personal growth and social development through extracurricular lectures, online popularization, and so on. It is also of necessity to create conditions for college students to participate in transformative tourism activities. For example, based on the selection of unique and meaningful tourism destinations, more study trips, practice week trips, on-campus club group trips, and off-campus club fellowship trips are expected.

*5.3. Research Shortcomings and Prospects*

Although this study has achieved certain innovations and breakthroughs, there are some limitations that need to be further explored in future studies. Firstly, according to the empirical results, the relationship between transformative tourism experiences and social connectedness needs to be explored in depth. Secondly, although this study confirms the influence and mechanism of transformative tourism experiences on college tourists' prosocial behaviors, it does not consider the possible moderating effects of many other influencing factors. As an illustration, travel motivation, life experience, personality traits, travel types, and cultural differences may all contribute subjectively or objectively to the differences in the effects of transformative tourism experiences on college tourists' prosocial behaviors. Future research can explore the moderating effect based on the above factors or conduct more detailed comparative studies. Finally, this study only uses questionnaires to collect data from Chinese college students. Future research might try to combine the investigation methods of questionnaires and experiments to improve the reliability and validity of the data, which can further test the causal relationship between transformative tourism experiences and college tourists' prosocial behaviors. Moreover, the sample selection of college students can be expanded to other countries so that the generalizability of the study results can be promoted. In addition, comparing the different representations of college tourists before and after experiencing transformative tourism and exploring their internal laws through a follow-up investigation is also one of the directions for further research.

## 6. Conclusions

Based on the SDT and combining the arousal conditions of individual dispositional awe and social connectedness, this study has constructed a multiple chain mediation model of "environmental stimuli (transformative tourism experiences)—emotional motivation (dispositional awe) & drive and intrinsic need motivations (social connectedness)—self-determined behaviors (prosocial behaviors)" with college tourists as the research object. The results have suggested that transformative tourism experiences can promote post-tour prosocial behaviors, and this effect is mediated by dispositional awe and social connectedness. The cultivation of college students' prosocial behaviors is of great significancesince it is linked to the healthy growth of individuals and the sound development of society. With tourism becoming a popular way of life and learning for college students, encouraging transformative tourism will be an important research topic in the field of prosocial behaviors of college students in the future, which will not only help to improve the efficiency of cultivating the long-lasting prosocial behaviors of college students, but will also have a positive effect on promoting the social education function of tourism and building a sustainable society.

**Author Contributions:** Conceptualization, Y.L., X.W. and Y.-J.C.; funding acquisition, Y.-J.G.; investigation, X.W.; project administration, Y.L.; supervision, Y.L., Y.-J.C. and Y.-J.G.; writing—original draft, X.W.; writing—review & editing, Y.L. and X.W. All authors have read and agreed to the published version of the manuscript.

**Funding:** This research was funded by Research on the construction of tourism destination governance system under the supply-side grant number 18BJY189.

**Institutional Review Board Statement:** Ethical review and approval were waived for this study due to the reason that this study is a retrospective study, and the questionnaire was used only to understand the subjects' previous transformative tourism experiences and their prosocial behaviors after the tour, without involving human biomedical research or patient privacy, and without any intervention on the subjects, and only to obtain and analyze the available data on the premise that the patients signed the subjects' informed consent and participated voluntarily.

**Informed Consent Statement:** Informed consent was obtained from all subjects involved in the study.Written informed consent has been obtained from the patient(s) to publish this paper.

**Data Availability Statement:** Not applicable.

**Conflicts of Interest:** The authors declare no conflict of interest.

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
