# Peer review of "The Impact of Transformative Tourism Experiences on Prosocial Behaviors of College Students: Multiple Chain Mediating Effects of Dispositional Awe and Social Connectedness"

_sustainability, doi:10.3390/su142013626_

Round 1
Reviewer 1 Report
The topic of the paper is very relevant. The methodology is clear and the limitations and implications are argued. Future research on the topic would be welcome.
Author Response
Dear reviewer,
Thank you for your approval. We will continue to improve the manuscript.
Kind regards,
The authors
Reviewer 2 Report
In reading this paper, I have more questions than this paper gives answers.
I think the authors need to spend a lot more time and effort selecting and understanding their participants prior to their undergoing "transformative tourism." I’m not sure a post-travel questionnaire is sufficient. I worry that the authors have merely taken results/assumptions from other tourist papers and are trying to see whether they fit Chinese demographics.
It seems especially risky to depend on people who self-identify as having had transformative experiences (the researchers should be able to qualify what counts as a transformative experience). What does this say about the participants? Perhaps they are ‘sheltered’ to begin with, just have a natural inclination to go biking, hiking and camping or maybe those who opted not to have a transformative experiences and were thus not surveyed couldn’t be lured to do so (supply isn’t sufficient to create demand). Furthermore, people who are naturally curious about biking and camping are less likely to be transformed, even if their vacation counts as “transformative tourism.” They are just doing what they like best.
I think more must be done to test whether people adopted pro social attitudes as a result of some transformative experience, or something else. In fact, the kinds of questions listed in Table 3 responses suggest that participants suffered a trauma and are really just happy to be home again. In fact H1 and H2 outcomes like "disposition awe" and "prosocial behaviors" exemplify this worry.
Moreover, the study's focus on college students seems to focus on the elite 16% whose interest in transformative experiences such as volunteer tourism, backpacker tourism, cycling tourism, dark tourism, red tourism, etc. is likely short-lived, since I doubt such activities will interest them once they’re in their 30s and have families and/or make a lot of money.
One question is, are the qualities of “open-mindedness, courage and adventuresomeness predispositions, independent particular to participants or are these characteristics actually outcomes of transformative tourism, as in previously unadventurous people become more adventurous. This is neither addressed nor proven in any meaningful way.
Finally, the authors appeal to Self-determination Theory (SDT), but I’m not sure that this theory is cross-cultural, let alone cross-generational. It seems perfect for well-to-do students who have enough confidence and sufficient resources to experience something out of the ordinary. All of the tests and questions seem like a lot of effort to make people more tolerant and build pro-social attitudes. Why not focus on Buddhism or Confucianism…it costs less and has the same overall outcome and is surely long-lasting.
There seems to be one element that is really missing…most people who experience transformative experiences tend to connect largely with people like themselves, and consider themselves superior to those who lacked similar opportunities when they were young. Unless one really had a traumatic experience, I don’t see how either H1 or H2 would be true in general. If participants are truly happy to be home, this rather defeats the point of “transformative experiences,” unless the goal is to improve people’s contentment so they don’t long for opportunities that they don't have.
Sometimes the paper reads like it is a research proposal, not the outcome of actual research.
I thought everyone in China is from a small family…how does this make Generation Z special?
Author Response
Dear reviewer,
Please see the enclosed Responses for specific replies.
Kind regards,
The authors

Reviewer 3 Report
The reviewed article presents interesting and up-to-date topic of the impact of tourism experiences on prosocial behaviors . The methodology of the study is properly desinged and conducted. And the research itself id is well established in the literature on the subjec. The paper is also well structured and the study results explained coherently. All requirements of the journal are met. The paper corresponds with the special issue topic.
Author Response

(The authors gave the same response as above.)

Round 2
Reviewer 2 Report
Line 403- I think you want to add "1" to hypothesis in that sentence.
Author Response
Dear reviewer,
Thank you for your comments and suggestions. We will add "1" to hypothesis in line 403 and check the English spelling.
Kind regards,
The authors